# Inhibition of NADPH Oxidases Prevents the Development of Osteoarthritis

**DOI:** 10.3390/antiox11122346

**Published:** 2022-11-27

**Authors:** Jin Han, Donghwi Park, Ji Young Park, Seungwoo Han

**Affiliations:** 1Laboratory for Arthritis and Cartilage Biology, Research Institute of Aging and Metabolism, Kyungpook National University, Daegu 41404, Republic of Korea; 2Department of Rehabilitation Medicine, School of Medicine, Ulsan University, Ulsan 44033, Republic of Korea; 3Departments of Pathology, School of Medicine, Kyungpook National University Hospital, 807, Hoguk-Ro, Buk-Gu, Daegu 41404, Republic of Korea; 4Division of Rheumatology, Department of Internal Medicine, School of Medicine, Kyungpook National University, 807, Hoguk-Ro, Buk-Gu, Daegu 41404, Republic of Korea

**Keywords:** osteoarthritis, NADPH oxidase, oxidative stress, reactive oxygen species, chondrocyte

## Abstract

Increased oxidative stress in osteoarthritis (OA) cartilage mediates catabolic signal transduction leading to extracellular matrix degradation and chondrocyte apoptosis. This study aimed to explore the contribution of NADPH oxidase (NOX), a major source of cellular reactive oxygen species (ROS), to the catabolic process of chondrocytes and to OA. The inhibition of NOX isoforms with a pan-NOX inhibitor, APX-115, significantly decreased IL-1β-induced ROS production in primary chondrocytes and, most potently, suppressed the expression of oxidative stress marker genes and catabolic proteases compared with the inhibition of other ROS sources. Catabolic stimuli by IL-1β treatment and in post-traumatic OA conditions upregulated the expression of NOX2 and NOX4 in chondrocytes. In the post-traumatic OA model, the pharmacologic inhibition of NOX protected mice against OA by modulating the oxidative stress and the expression of MMP-13 and Adamts5 in chondrocytes. Mechanistically, NOX inhibition suppresses Rac1, p38, and JNK MAPK signaling consistently and restores oxidative phosphorylation in IL-1β-treated chondrocytes. In conclusion, NOX inhibition prevented the development of OA by attenuating the catabolic signaling and restoring the mitochondrial metabolism and can thus be a promising class of drug for OA.

## 1. Introduction

Oxidative stress, which occurs when the production of reactive oxygen species (ROS) exceeds antioxidant capacities, has been known to be causally important in the development of osteoarthritis (OA) [1,2]. Excessive ROS as signaling intermediates can activate redox-sensitive kinases, including c-Src and apoptosis signal-regulating kinase 1 (ASK1) [3,4]. It amplifies diverse signaling cascades elicited by cytokines, growth factors, and extracellular matrix (ECM) proteins, consequently increasing catabolic proteases, such as matrix metallopeptidase-13 (MMP-13) and a disintegrin and metalloproteinase with thrombospondin motifs 5 (Adamts5) in chondrocytes [4,5]. In addition, ROS can lead to mitochondrial dysfunction, senescence, and, eventually, chondrocyte apoptosis [6]. The majority of intracellular ROS are generated by mitochondria in response to various stimuli, such as serum deprivation, hypoxia, integrin signaling, inflammatory cytokines, and apoptotic signaling [7,8]. Aside from the mitochondrial ROS system, other ROS-producing enzyme systems include xanthine oxidase, lipid peroxidases, cytochrome P450 enzymes, and nitric oxide (NO) synthase [7,9]. However, ROS production itself is not the primary purpose of these enzymes but is a byproduct of their enzymatic reaction. The only enzyme that primarily functions to produce ROS is NADPH oxidase (NOX) [10].

NOX is a membrane-bound enzyme complex that produces ROS by transferring one electron to oxygen from NADPH, an essential electron donor in all organisms [10]. The human NOX system is composed of NOX1 to NOX5, dual oxidase (DUOX)1, and DUOX2. NOX-derived ROS are functionally essential for bacterial killing and innate immunity, but excessive ROS production can lead to tissue damage and a prolonged inflammatory response [11]. Therefore, the NOX system is dormant in resting cells and becomes activated when it is assembled into a complex at the cytoplasmic membrane by the binding of membrane-bound proteins, such as gp91^phox^ and p22^phox^, to cytosolic proteins, p67^phox^, p47^phox^, p40^phox^, and Rac. Although the activation mechanism of the NOX system in the situation of OA is not fully understood, possible candidates include the senescence-associated secretory phenotype (SASP), including interleukin (IL)-1β, IL-6, and heme oxygenase-1, and the damage-associated molecular patterns (DAMPs) released during cartilage degradation, including cartilage compositions, calcium crystals, and endogenous molecules [12]. Among NOX isoforms, NOX1, NOX2, and NOX4 are mainly expressed in chondrocytes. Conversely, NOX3 is only slightly expressed, and NOX5, DUOX1, and DUOX2 are not [13]. Compared with NOX1 and NOX2, which are expressed on the cell surface and in prehypertrophic and hypertrophic chondrocytes, NOX4 is primarily expressed in the endoplasmic reticulum and mitochondria of proliferating and prehypertrophic chondrocytes [13]. During endochondral bone formation, the depletion of NOX2 and NOX4 suppresses ROS generation in chondrocytes, leading to the inhibition of differentiation and proteoglycan production and eventually to the apoptosis of chondrocytes [13]. In human OA chondrocytes, NOX4 is primarily expressed among the NOX isoforms and is involved in the production of MMP-1, MMP-13, and Adamts4 by IL-1β [14]. However, the contribution of NOX-mediated ROS production to the development of OA is not well understood.

Our group previously reported that extracellular matrix (ECM) degradation can reproduce the catabolic OA phenotype, such as hypertrophy-like morphological changes and increased MMP-13 and ADAMTS5 expression in chondrocytes [15]. Interestingly, the breakdown of ECM consistently increased ROS production and oxidative stress-marker genes, including *Fth1*, *Hmox1*, and *Txn*. However, it is suppressed by inhibiting Toll-like receptor (TLR) 2 or TLR4 [15]. Based on these data, DAMPs generated during ECM breakdown can thus trigger TLR signaling, which produces an oxidative stress-related signature in chondrocytes. We hypothesized that the NOX system in the cell membrane may be responsible for the enhanced oxidative stress and catabolic process associated with ECM breakdown. In this study, we investigated the relative role of the NOX system among cellular ROS production systems in IL-1β-induced catabolic conditions in chondrocytes. The expression of NOX isoforms and the pharmacologic NOX inhibition phenotype were then assessed in vitro in IL-1β-treated chondrocytes and in vivo in a surgical OA model. Finally, to investigate the molecular mechanism of NOX-mediated catabolic effects, we analyzed the changes in redox-sensitive kinase-related signaling and in mitochondrial metabolism in primary chondrocytes.

## 2. Materials and Methods

### 2.1. Animal Experiments and Ethics

The experimental OA was induced in 12 week old male C57BL/6 mice by destabilization of the medial meniscus (DMM) surgery under general anesthesia using a standard protocol, which was conducted with 7 mice per group, and all data on the 7 mice were used in the analysis. This sample size was determined based on a previous study in which the desired effect sizes were shown to be statistically significant [16]. The animals were randomly allocated into the treatment group and control group, which was awarded by J. H. and D. P. during the experiment and data analysis. The treatment group was injected intraperitoneally with APX-115 at 60 mg/kg, whereas the control group was injected with the same volume of 5% DMSO twice a week. The dose of APX-115 was determined by a previous study, which showed clinically significant anti-inflammatory effects in diabetic mice [16]. Briefly, the bioavailability of the small molecules measured by the area under the curve (AUC) revealed to be 1.5- to 6-fold higher after intraperitoneal administration compared to oral administration [17]. We assumed that the bioavailability of IP was approximately 3 times higher than the oral intake and could obtain a similar AUC with oral intake if APX-115 was given intraperitoneally twice a week. To deliver 60 mg/kg of APX-115 to 30 g weight mice, 18 µL of APX-115 stock solution (100 mg/mL) was mixed with normal saline 342 µL and intraperitoneally injected. The order of the injection was randomly decided, and the mice were raised under the same conditions to minimize potential confounders. The knee joints were harvested for histological analysis 8 weeks after surgery. The C57BL/6 mice were cared for under pathogen-free conditions. All experiments were conducted in accordance with approved animal protocols and guidelines established by the Animal Care Committee of Kyungpook National University (approval no: KNU-2022-0203). This study was carried out in accordance with the recommendations in the ARRIVE 2.0 guidelines [18].

### 2.2. Primary Chondrocyte Cultures

The primary chondrocytes were obtained from the long bones of embryonic day 15.5 C57BL/6 mice based on the evidence that the long bone of C57BL/6J mice is avascular and is mainly composed of resting and proliferative chondrocytes at peripheral areas and of hypertrophic chondrocytes at the primary ossification center at embryonic day 15.5 [19]. Briefly, the isolated bones were equilibrated in α-minimum essential medium (α-MEM)-based organ culture media supplemented with 0.2% BSA, 0.25 mM ascorbic acid, 1 mM β-glycerophosphate, 0.25% L-glutamine, and 0.25% penicillin/streptomycin at 37 °C in a 5% CO_2_ for 24 h. They were then incubated in 0.25% trypsin-EDTA with gentle shaking for 15 min at 37 °C and digested with 3 mg/mL of collagenase P (Sigma-Aldrich, St. Louis, MO, USA) in complete Dulbecco’s modified essential media (DMEM) for 2 h. The fractioned cells were filtered through a 40 µm nylon mesh (BD Bioscience, Franklin lakes, NJ, USA) and collected by centrifugation. A cell culture was performed in 3:2 F12:DMEM media containing 10% fetal bovine serum (FBS) and 0.25% L-glutamine. The obtained chondrocytes were seeded in 6-well plates at a density of 500,000 cells/well and cultured in 2:3 DMEM:F12 medium supplemented with 10% FBS, 0.25% L-glutamine, and 0.25% penicillin/streptomycin. To determine the effect of the NOX isoforms, NOX1 inhibitor (ML171; 10 µM), NOX2 inhibitor (GSK2795039; 10 µM), NOX4 inhibitor (GLX351322; 10 µM), and pan-NOX inhibitor (APX-115) were pretreated 30 min before the IL-1β (10 ng/mL) treatment. For the signaling analysis, starved chondrocytes were pretreated for 30 min with 10 µM pan-NOX inhibitor, APX-115, or cotreated with 10 µM NOX2 inhibitor (GSK2795039) and NOX4 inhibitor (GLX351322) and then stimulated with 20 ng/mL IL-1β. Primary chondrocytes at passage 0 were only used for the experiments to minimize the dedifferentiation of chondrocytes [20].

### 2.3. Detection of Intracellular ROS and Oxidative DNA Damage

Intracellular ROS production in the chondrocytes were detected with the cell-permeable fluorescent dye dihydroethidium (DHE), and oxidative DNA damage was determined with 8-oxo-dG antibody, the oxidized derivative of deoxyguanosine. The primary chondrocytes were treated with IL-1β (10 ng/mL) with or without 2-deoxy-d-glucose (2-DG; 2 mM), IACS-010759 (2 µM), 1400W (10 µM), and APX-115 (10 µM) for 24 h. The cells were fixed with 4% paraformaldehyde for 10 min, permeabilized with 0.25% Triton X-100, and washed with phosphate-buffered saline (PBS) three times. To assess intracellular ROS production, 5 µM DHE was added for 30 min at 37 °C. The cells were washed with PBS, and photos of the cells were taken with an KI-3000F fluorescence microscope (Korealabtech, Pyeongtaek, Republic of Korea). For the immunofluorescence staining of 8-oxo-dG, the cells were incubated with primary antibody for 8-oxo-dG (sc-66036; Santa Cruz Technology, Beverly, MA, USA) overnight at 4 °C and incubated with a secondary antibody for 2 h and then photographed under the fluorescence microscope. Fluorescence-positive cells were quantified with ImageJ software (version 1.8.0, National Institutes of Health, Bethesda, MD, USA).

### 2.4. Quantitative Real-Time RT-PCR

The total RNA was isolated from primary chondrocyte cultures using TRIzol (Invitrogen, Carlsbad, CA, USA), and the first strand cDNA was synthesized using Superscript III reverse transcriptase (Invitrogen). Real-time qPCR was performed using a ViiA™ 7 Real-Time PCR System (Applied Biosystems, Foster City, CA, USA) and SYBR^®^ Green Master Mix (Applied Biosystems). The primers used for real-time qPCR are listed in Appendix A. All the RT-qPCR reactions were performed in triplicate and repeated two to three times. Among them, the representative results are shown. The expression levels of the target genes were normalized to the geometric mean of *Gapdh* as an internal control and calculated using the 2^−ΔΔCT^ method. A list of the primers used for the RT-qPCR are shown in Appendix A.

### 2.5. Western Blot Analysis

The proteins from the primary chondrocyte cultures (*n* = 3 independent isolations) were harvested using 300 µL RIPA buffer supplemented with protease and phosphatase inhibitors (Roche Diagnostics, Indianapolis, IN). The total cell lysates (containing 20–30 µg protein) were separated by 10% SDS-polyacrylamide gel electrophoresis and transferred to polyvinylidene difluoride membranes (Immobilon-P; Millipore Corporation, Billerica, MA, USA). The membranes were then blocked using 5% skimmed milk in PBS with 0.25% Tween-20 (PBST) and incubated at 4 °C overnight with the primary antibodies listed in Appendix A. After washing with PBST, the membranes were bound to horseradish peroxidase-conjugated secondary antibodies and incubated at RT for 2 h. After washing with PBST, the blots were developed using enhanced chemiluminescence Western blotting detection reagent (Thermo Fisher Scientific, Waltham, MA, USA) and analyzed with the MicroChemi system (DNR Bio-imaging Systems, Neve Yamin, Israel). The Western blot band intensities were quantified with ImageJ software. A list of the primary antibodies used for the Western blot analysis are shown in Appendix A.

### 2.6. Safranin-O Staining, Immunofluorescence, and Immunohistochemistry of Mouse Knee Joint

Mouse knee joints were fixed in 4% paraformaldehyde for 24 h, decalcified in 10% EDTA for 3 weeks, and embedded in paraffin. The embedded blocks were sectioned at a thickness of 6 µm and stained with Safranin-O/Fast Green. The cartilage destruction in all four quadrants of the joint (grade 0–24) and medial tibial plateau (grade 0–6) was scored by two observers under blinded conditions using the OARSI score system [21]. For immunofluorescent and immunohistochemical staining, the rehydrated sections were retrieved in sodium citrate buffer (10 mM sodium citrate, 0.05% Tween 20, and pH 6.0). After the sections were blocked with 2% bovine serum albumin (BSA) in PBS for 1 h, they were incubated with the primary antibodies for NOX1, NOX2, NOX3, NOX4, MMP13, 8-oxo-dG, or normal rabbit IgG (Appendix A) in 1% BSA at 4 °C overnight. For immunofluorescence staining, the sections were incubated with Alexa Fluor 488- or 594-conjugated secondary antibodies (Jackson Immuno Research Laboratories, West Grove, PA, USA) for 2 h and counterstained with DAPI. For immunohistochemistry, the sections were incubated with a biotinylated anti-rabbit secondary antibody (BA-1000; Sigma-Aldrich) and visualized with the Vector Red Alkaline Phosphatase Kit (AK-5200, Vector Labs, Burlingame, CA, USA). The sections were mounted with an anti-fade mounting solution (Vector Labs) and imaged and quantified under a KI-3000F fluorescence microscope. The immunostaining images of the NOX isoforms, MMP13, and 8-oxo-dG were quantified in the upper part of the tide mark of the lateral tibial plateau.

### 2.7. Extracellular Flux Analysis

The chondrocyte metabolism was analyzed with an XF96 Extracellular Flux Analyzer (Seahorse Bioscience Inc., Billerica, MA, USA). Briefly, primary chondrocytes were plated in Seahorse XF96 plates at a density of 50,000 cells/well. The confluent chondrocytes were preincubated with or without IL-1β (10 ng/mL) and with or without GSK2795039 (10 µM), GLX351322 (10 µM), and APX-155 for 24 h. For the Mito stress test, the cells were equilibrated with serum-free Seahorse XF Base Medium for 1 h. Then, sequential injections of 1 µM oligomycin, 1 µM FCCP (carbonyl cyanide-4-(trifluoromethoxy)phenylhydrazone), and 1:1 mixture of antimycin A (2 µM) with 1 µM rotenone were performed. Mitochondrial ATP production is expressed as the oxygen consumption rate (OCR; pmol of O_2_/min). For the glycolysis stress test, the cells were serum-starved for 1 h in glucose-free Seahorse XF DMEM and then sequentially treated with 20 mM glucose, 1 µM oligomycin, and 100 mM 2-DG with real-time measurements of proton accumulation in media, which was quantified as the extracellular acidification rate (ECAR; mpH/min) [22]. Once the Mito stress test or glycolysis stress test were completed, the nuclei of cells were counted after in situ DAPI staining, and the analysis data were normalized based on the cell count of each well with the normalization unit of 5000 cells. The data from the Seahorse analysis were from technical replicates, i.e., triplicate analyses on a single sample.

### 2.8. Statistical Analysis

All data are presented herein as the mean ± standard deviation (SD). A statistical analysis to compare the mean values of two groups was performed using the Mann–Whitney U test; a nonparametric test as the sample size is small and it cannot be assumed as a normal distribution. *p*-Values ≤ 0.05 were considered statistically significant. Statistical analyses were performed with Prism software version 8.0 (GraphPad Software, La Jolla, CA, USA).

## 3. Results

### 3.1. NOX System Was Critically Involved in the IL-1β-Induced ROS Production and Oxidative DNA Damage in Chondrocytes

To elucidate which cellular mechanisms are involved in the IL-1β-induced ROS generation in the primary chondrocytes, IL-1β (10 ng/mL) was applied together with various inhibitors for 24 h. Glycolysis inhibitor (2-DG), oxidative phosphorylation (OxPhos) inhibitor (IACS-010759), inducible NOS inhibitor (1400 W), and pan-NOX inhibitor (APX-115) effectively decreased ROS generation as well as oxidative DNA damage in IL-1β-treated primary chondrocytes. Among the inhibitors, 2-DG, 1400 W, and APX-115 showed similar levels of inhibitory effects on ROS production and oxidative DNA damage (Figure 1A,B). We further assessed the expression of oxidative stress response genes, such as *Fth1*, *Txn*, and *Hmox*, under IL-1β treatment. Although all the inhibitors significantly suppressed the expression of *Fth1*, *Txn*, and *Hmox* in the stimulated chondrocytes, APX-115 was most effective (Figure 1C). The inhibitory effect of APX-115 was also observed on the expression of *MMP-13* and *Adamts5* in IL-1β-treated chondrocytes (Figure 1D).

### 3.2. NOX2 and NOX4 Was Upregulated in IL-1β-Treated and OA Chondrocytes

We then examined the expression of the NOX isoforms by IL-1β in primary chondrocytes. The IL-1β treatment significantly increased the mRNA expression of NOX2 at 6 and 12 h after treatment and that of NOX4 at 6 h but rather decreased those of NOX1 and NOX3 (Figure 2A). This pattern was reproduced in the Western blot analysis; IL-1β treatment profoundly increased the NOX2 protein levels in the primary chondrocytes and, to a lesser extent, that of NOX4. However, those of NOX1 and NOX3 were significantly decreased by IL-1β (Figure 2B,C). Immunofluorescence staining for the NOX isoforms in DMM-induced OA mice also revealed a gradual increases in NOX2 and NOX4 but decreases in NOX1 and NOX3 in the articular cartilages (Figure 2D,E and Appendix A). These in vitro and in vivo data suggest that the NOX isoforms have different expression patterns under catabolic conditions of OA compared to physiologic conditions.

### 3.3. NOX Inhibition Attenuated OA Severity by Modulating Oxidative Damage and Expression of MMP-13 and Adamts5 in Chondrocytes

To evaluate the effect of the inhibition of the NOX isoforms on cartilage damage in surgically induced OA, APX-115 was intraperitoneally injected twice a week in DMM-induced OA mice. The treatment of APX-115 significantly reduced the severity of cartilage damage compared with control DMM mice. The decreased severity of OA was accompanied by a decrease in 8-oxo-dG, a marker of oxidative DNA damage, as well as MMP-13 expression in the articular chondrocytes (Figure 3A,B, Appendix A). The in vitro functions of NOX inhibition were further examined in IL-1β-treated primary chondrocytes. The inhibition of NOX2 with GSK2795039 decreased the mRNA expression of MMP-13 and Adamts5 but failed to affect their protein level. The NOX4 inhibitor GLX351322 significantly decreased the protein level of MMP-13 and Adamts5. Cotreatment of both the NOX2 and NOX4 inhibitors significantly decreased both MMP-13 and Adamts5 and increased Col2 and Aggrecan in mRNA and protein levels under IL-1β-treated catabolic condition. In addition, pan-NOX inhibitor, APX-115 markedly suppressed MMP-13 and Adamts5 but considerably increased those of Col2 and Aggrecan in mRNA and protein levels (Figure 3C,D). These anabolic effects of the NOX inhibitors were accompanied by the decrease in oxidative stress-target genes, such as *Fth* and *Txn* (Figure 3C). These results suggest that the inhibition of NOX isoforms can suppress the catabolic processes by IL-1β and enhance anabolic processes in chondrocytes, thereby conferring a protective effect in the surgical OA model.

### 3.4. NOX Inhibition Consistently Suppressed Rac1, p38, and JNK MAPK Signaling in IL-1β-Treated Chondrocytes

To elucidate the molecular mechanisms of NOX inhibition associated with the protective effects on chondrocytes, we further examined the function of the NOX isoforms in IL-1β-mediated signaling activation, because ROS amplify inflammatory signaling, particularly with respect to MAPK and NF-κB signaling [3]. Pan-NOX inhibition with APX-115 markedly suppressed the activated form of Rac1 (Rac1-GTP) and the phosphorylation of p38 and JNK MAPK but not ERK1/2. In NF-kB signaling, the phosphorylation of IkB was decreased by APX-115 at 5 min, but not p65. In addition, the phosphorylation of AKT and STAT3 was not affected by APX-115 (Figure 4A).

To narrow down the effect of pan-NOX inhibitor on catabolic signaling by IL-1β, we assessed MAPK and NF-κB signaling in the presence of both NOX2 inhibitor (GSK2795039) and NOX4 inhibitor (GLX351322). The inhibition of both NOX2 and NOX4 reduced the activation of Rac1, p38, and JNK MAPK as in the pan-NOX inhibitor. The phosphorylation of IkB was suppressed by the inhibition of NOX2 and NOX4, but that of p65, AKT, and STAT3 was not affected as in the case of pan-NOX inhibition (Figure 4B).

### 3.5. NOX Inhibition Restored Mitochondrial Respiration in IL-1β-Treated Chondrocytes

We showed that NOX inhibition suppressed most effectively the oxidative stress-marker genes and catabolic proteases, such as MMP-13 and Adamts5 by IL-1β (Figure 1). Considering that mitochondria are the major source of ROS produced by inflammatory cytokines [8], our findings suggest the regulation of mitochondrial respiration by the NOX system in chondrocytes. To confirm this, we assessed whether NOX inhibition affected OxPhos and glycolysis under IL-1β-induced catabolic conditions using the Seahorse XF-96 Extracellular Flux Analyzer. In the Mito stress test, IL-1β treatment for 24 h markedly decreased basal and maximal respirations of the cultured primary chondrocytes. Interestingly, the inhibition of both NOX2 and NOX4 most effectively restored the decrease in mitochondrial respiration by IL-1β, followed by pan-NOX, NOX2, and NOX4 inhibition, which all showed a similar degree of recovery in chondrocyte mitochondrial basal and maximal respiration (Figure 5A). The ECAR analysis performed to assess glycolysis revealed that the primary chondrocytes treated with IL-1β had a higher glycolytic capacity. The inhibition of NOX2 or NOX4 and pan-NOX partially normalized the increased glycolysis and glycolytic capacity induced by IL-1β in the chondrocytes (Figure 5B). These findings suggest that NOX inhibition can restore mitochondrial respiration and metabolic dysfunction in OA chondrocytes.

## 4. Discussion

Under oxidative stress conditions in OA, excessive ROS can foster aberrant activation of intracellular signaling, senescence, and chondrocyte apoptosis. The present study showed that the NOX system is critically involved in IL-1β-induced ROS production and oxidative stress in chondrocytes. Among the NOX isoforms, NOX2 and NOX4 were upregulated in IL-1β-treated and OA chondrocytes. The pharmacologic inhibition of NOX isoforms prevents the development of OA by reducing oxidative stress as well as catabolic proteases, such as MMP-13 and Adamts5. Mechanistically, NOX-dependent ROS production is a critical regulator of Rac1, p38, and JNK MAPK signaling and of mitochondrial metabolism in IL-1β-treated chondrocytes.

Interestingly, we found that the inhibition of NOX isoforms most potently suppressed the expression of oxidative stress marker genes and catabolic proteases compared with the inhibition of other ROS production systems in chondrocytes (Figure 1). Although mitochondria are believed to be the main source of ROS in cells, direct evidence supporting this idea is rare [7]. A recently developed cell-based assay for detecting the oxidation of a fluorogenic dye revealed that mitochondria are never the largest contributor of cellular ROS in several cell types [23]. When dissected for the source of the cellular ROS, mitochondria contributed less than 50% in all of the tested cell types. Instead, other cellular enzyme systems, especially NOX, have been identified as the main sources of ROS production in both resting and stressed cells [23].

A novel finding of this study is the suppression of IL-1β-mediated Rac1, p38, and JNK MAPK signaling by both pan-NOX inhibitor and NOX2/4 dual inhibition. The Rho-like small GTPases, Rac1 is known to be critical for the activation of NOX isoforms [24]. Rac1 is ubiquitously expressed and activates NOX in nonhematopoietic cells, while Rac2 expression is confined to hematopoietic cells, specifically neutrophils [24,25]. Activated Rac1 recruits p67phox, which associates with p47phox and cytochrome leading to the activation of NOX complex [26]. Our data suggest that Rac1 is not only involved in NOX activation, but also its activity is regulated by ROS produced by NOX, forming a positive feedback loop. This Rac1-NOX-ROS loop could likely play an important role in cellular damage associated with excessive ROS production, but its biological meaning is still not understood. Next, p38 and JNK MAPK is well known as a downstream target of redox-sensitive kinases. The H_2_O_2_ generated by the NOX-receptor kinase interaction can amplify the signal transduction by activation of redox-sensitive kinases, such as apoptosis signal-regulating kinase 1 (ASK1), MAP three kinase 1 (MTK1), and c-Src [3,4,27]. ASK1 is a MAP3K that is responsible for activating p38 and JNK MAPK, and normally it binds to the inhibitory protein, thioredoxin (Trx1) [3]. The H2O2-mediated oxidation dissociates Trx1 from ASK1, allowing oligomerization and activation of ASK1 to initiate downstream signaling [28]. Recently, MTK1 was also shown to function as an oxidative-stress sensor that induces a stress-activated p38 and JNK MAPK signaling [27]. The p38 and JNK MAPK signaling is known to participate in the pathogenesis of OA through the induction of cellular senescence, the hypertrophic differentiation, the synthesis of MMPs, and the production of pro-inflammatory factors [29]. Our data suggest that the NOX system, as a major source of cellular ROS, plays an important role in the activation of p38 and JNK MAPK signaling during OA progression.

Among the NOX isoforms, NOX2 and NOX4 was predominantly expressed in OA chondrocytes in vivo and in IL-1β-treated chondrocytes in vitro, indicating them as key sources of ROS during cartilage degradation. NOX2 is composed of two membrane subunits, the catalytic subunit, gp91phox (“phox” stands for phagocytic oxidase), p22phox, and three cytoplasmic regulatory subunits: p40phox, p47phox, and p67phox. NOX4 forms a heterodimer with p22phox without a cytosolic subunit. Several studies have consistently reported the critical roles of NOX2 and NOX4 in OA [30]. NOX2 is weakly expressed in normal chondrocytes but increases in OA and IL-1β-treated chondrocytes [30,31]. In the collagenase-induced OA model, NOX2 depletion showed a significant decrease in OA severity as well as synovial lining proliferation and osteophyte formation [32]. In addition, the inhibition of p47phox, the cytosolic subunit of NOX2, using siRNA-loaded poly(lactic-co-glycolic) acid nanoparticles decreased ROS production and, consequently, attenuated pain behaviors, and cartilage damage in mono-iodoacetate–induced OA [33]. NOX4 is elevated in OA chondrocytes, and its expression is enhanced by IL-1β treatment in human articular chondrocytes [14]. In a traumatic OA model, NOX4 expression increased within 48 h in chondrocytes and persisted for 7 days [34]. Functionally, NOX4 is critical for the expression of MMP-13 and Adamts5 and the production of advanced oxidation protein products, which are positively associated with OA severity [14,35,36]. On the other hand, NOX1 and NOX3 was downregulated by IL-1β treatment in vitro and in chondrocytes of DMM model in vivo. The expression of NOX1 and NOX3 in unstimulated primary chondrocytes suggests that they play a role in ROS production of physiologic condition such as endochondral bone formation [37]. Further study is needed to elucidate the role of NOX1 and NOX3 during endochondral bone formation and differentiation of mesenchymal cells.

Besides the putative role in ROS production, our results revealed NOX-dependent regulation of the mitochondrial metabolism, which can contribute to mitochondrial ROS production. In fact, treatment with the pan-NOX inhibitor APX-115 or dual inhibition of NOX2 and NOX4 restored the IL-1β-mediated impairment of OxPhos and enhancement of glycolysis. One possible explanation is a direct regulation of mitochondrial OxPhos by NOX-mediated ROS production [38]. The inhibition of NOX isoforms directly suppresses a cellular ROS production, and it can be responsible for the restoration of IL-1β-mediated metabolic shifting to glycolysis in chondrocytes. In addition, our data showed that NOX inhibition suppressed the activation of p38 and JNK MAPK by IL-1β. The evidence showed that sustained activation of p38 and JNK MAPK signaling can increase mitochondrial respiration as well as ROS production through p38α-activated kinase, MAPKAPK-2 [39], and through direct phosphorylation of pyruvate kinase (PK) M2 and pyruvate dehydrogenase (PDH) by JNK [40]. Although less than 10% of the total cellular ATP is derived from OxPhos, OA chondrocytes have shown an increasing trend of OxPhos-induced ATP production compared with normal chondrocytes [41]. This metabolic change is believed to be an adaptation of chondrocytes to stressful OA conditions. However, persistent stress, along with the activation of inflammatory signaling cascades, such as the p38 and JNK MAPK pathways, can lead to metabolic compensation failure and, eventually, to chondrocyte senescence and apoptosis [40,42,43,44]. Our data suggest that the NOX system may play a critical role during these processes in the dysregulation of mitochondrial metabolism, directly through ROS production and indirectly through the regulation of the inflammatory signaling cascade (Figure 6).

## 5. Conclusions

Our findings provide insight into how ROS produced by the NOX system affects cartilage degradation in OA. We found that the NOX system is critical in the ROS production and oxidative stress induced by IL-1β and in the development of surgically induced OA. Its catabolic effects works through the amplification of Rac1 and p38 MAPK signaling and the dysregulation of mitochondrial metabolism (Figure 6). Further research is needed to delineate the role of NOX2 and NOX4 in OA and to validate a more specific NOX2 or NOX4 inhibitor as a therapeutic target in OA.

## Figures and Tables

**Figure 1 antioxidants-11-02346-f001:**
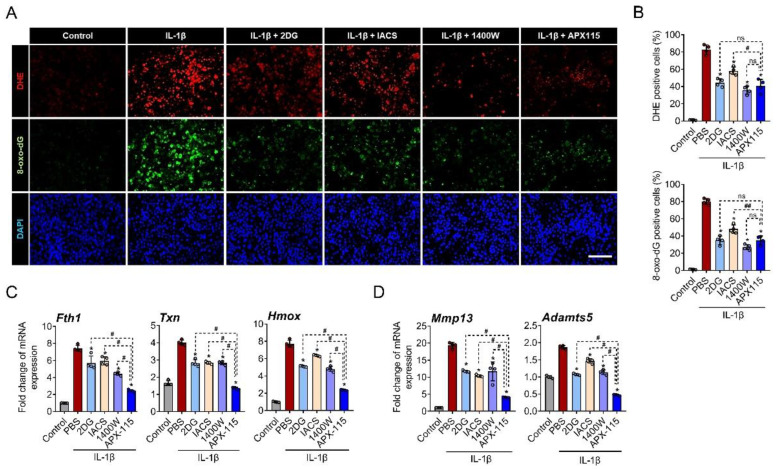
NADPH oxidase (NOX) is critically involved in IL-1β-induced oxidative stress. (**A**) Primary chondrocytes treated with IL-1β (10 ng/mL) along with 2-DG (glycolysis inhibitor), IACS010759 (oxidative phosphorylation inhibitor), 1400 W (iNOS inhibitor), and APX-115 (pan-NOX inhibitor) for 24 h were stained with dihydroethidium (DHE; red) as a fluorescent probe for ROS and immunostained with 8-oxo-dG (green) as a marker for oxidative DNA damage. Nuclei of cells were counterstained with 4′,6-diamidino-2-phenylindole (DAPI; blue). Scale bar indicates 100 µm. (**B**) DHE and 8-oxo-dG fluorescence-positive cells were quantified and expressed as the ratio to DAPI-positive cells. (**C**) Primary chondrocytes were treated with IL-1β and inhibitors for 12 h, and the relative expression of oxidative stress marker genes, *Fth1*, *Hmox*, and *Txn* mRNA, was evaluated by real-time reverse transcription polymerase chain reaction. mRNA expression level is presented as the fold increase, which was normalized to a *Gapdh* value. (**D**) The relative mRNA expression of *MMP-13* and *Adamts5* is presented as the fold increase relative to *Gapdh* expression. (**B**–**D**) * *p* < 0.05, compared with the IL-1β-treated group (red bar); ^#^
*p* < 0.05 and ^##^
*p* < 0.001. Mann–Whitney U test, *n* = 4; ns, not significant.

**Figure 2 antioxidants-11-02346-f002:**
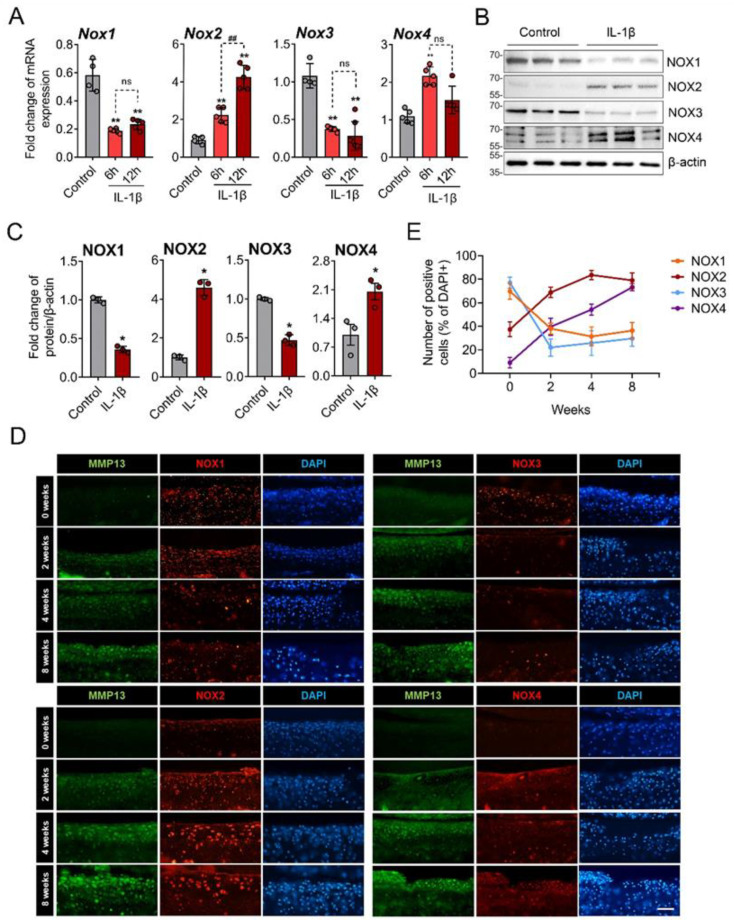
NOX2 and NOX4 were increased by IL-1β in vitro and in surgically induced OA cartilage. (**A**) Primary chondrocytes were treated with IL-1β for 6 and 12 h, and mRNAs were subjected to real-time reverse transcription polymerase chain reaction. The relative mRNA expression of NOX family genes, including *NOX1*, *NOX2*, *NOX3*, and *NOX4*, were quantified. * *p* < 0.05 and ** *p* < 0.001, compared with the control group (grey bar); ^##^
*p* < 0.001. Mann–Whitney U test, *n* = 4; ns, not significant. (**B**) Western blot images for the NOX isoforms in primary chondrocytes treated with or without IL-1β (10 ng/mL) for 24 h. (**C**) The Western blot band intensities were quantified by densitometric analysis using the ImageJ program and normalized to that of β-actin, which is displayed as relative densitometric bar graphs. * *p* < 0.05 compared to the control group (grey bar), Mann–Whitney U test, *n* = 3. (**D**) Immunofluorescence staining for the NOX isoforms (red) in knee cartilage from the destabilization of the medial meniscus (DMM)-induced OA mice at 0, 2, 4, and 8 weeks. MMP-13 (green) was used as positive control for OA progression in each sample (*n* = 5). Scale bar indicates 100 µm. (**E**) The percentage of NOX-expressing cells over DAPI-positive cells was quantified at 0, 2, 4, and 8 weeks after DMM surgery and expressed as the mean ± SD.

**Figure 3 antioxidants-11-02346-f003:**
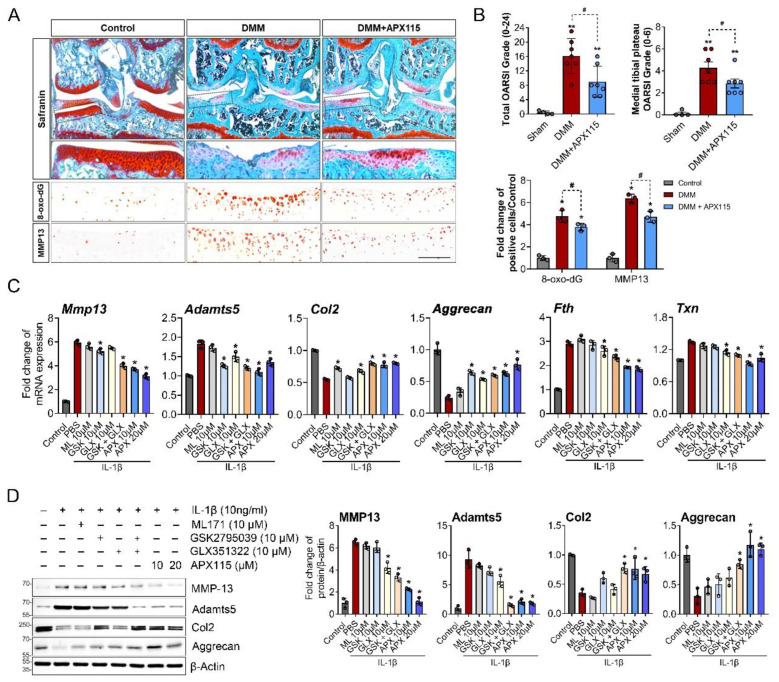
NOX inhibition attenuated cartilage damage and oxidative stress in the chondrocytes in surgically induced OA. (**A**) Representative images of Safranin-O staining and immunostaining for MMP-13 and 8-oxo-dG. DMM surgery was conducted in 12 week old male C57BL/6 mice, and APX-115, a pan-NOX inhibitor, was administered intraperitoneally at 60 mg/kg twice a week for 8 weeks. Scale bar indicates 100 µm. The source data are provided in Appendix A. (**B**) Total and medial tibial plateau Osteoarthritis Research Society International (OARSI) grade and the relative numbers of MMP-13- and 8-oxo-dG-positive cells were quantified and expressed as the mean ± SD. * *p* < 0.05 and ** *p* < 0.001, compared with the sham group (grey bar); ^#^
*p* < 0.05, compared with the DMM group (red bar). Mann–Whitney U test, *n* = 4 for the sham and *n* = 7 for the DMM and DMM + APX115 groups. (**C**) Relative mRNA levels of *MMP-13*, *Adamts5*, *Col2*, *Aggrecan,* and oxidative stress target genes of *Fth* and *Txn* in the primary chondrocytes treated with IL-1β and NOX1 inhibitor; ML171 (10 µM), NOX2 inhibitor (GSK2795039) (10 µM), NOX4 inhibitor (GLX351322 (10 µM)), or pan-NOX inhibitor (APX-115 ((10 and 20 µM) for 6 h. (**D**) Representative Western blot image for MMP-13, Adamts5, Col2, and Aggrecan. IL-1β (10 ng/mL) was applied to primary chondrocytes with or without various doses of ML171, GSK2795039, GLX351322, or APX-115 for 24 h. The band density was quantified and normalized to that of β-actin, which is presented as the mean arbitrary density units from three biological replicates. (**C**,**D**) * *p* < 0.05 and ** *p* < 0.001 compared with the IL-1β-treated group (red bar). Mann–Whitney U test, *n* = 3.

**Figure 4 antioxidants-11-02346-f004:**
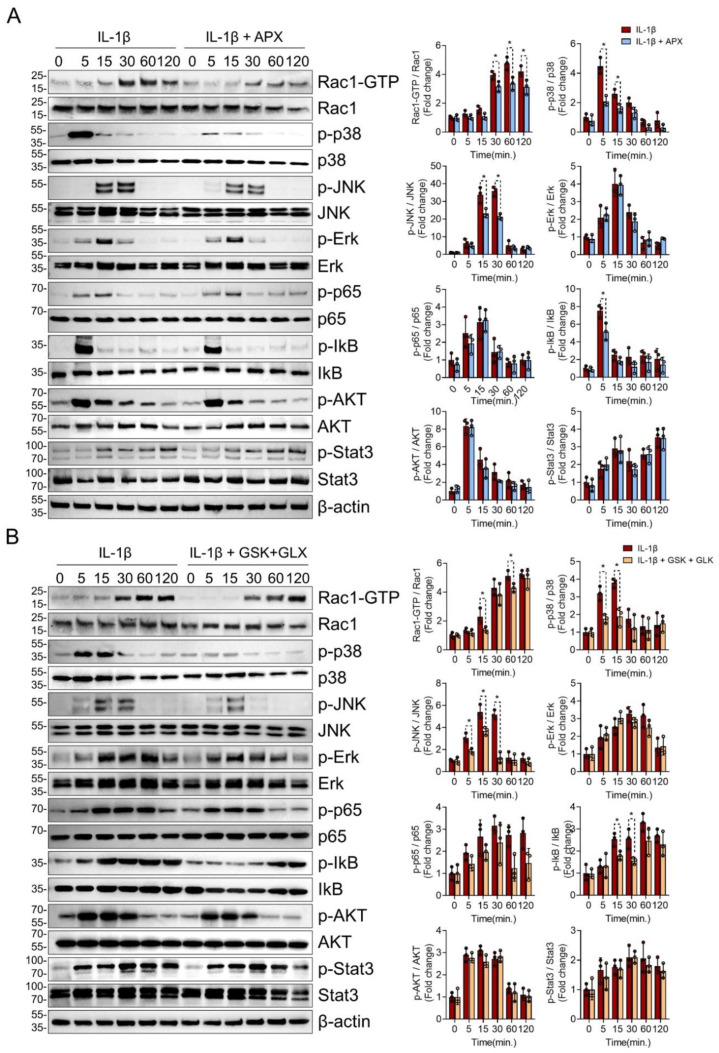
NOX inhibition attenuated Rac1, p38, and JNK MAPK signaling in IL-1β-treated chondrocytes. (**A**) Effects of pan-NOX inhibition on chondrocyte catabolic signaling activated in response to IL-1β. Starved primary chondrocytes with or without APX-115 (10 µM) pretreatment were stimulated by IL-1β (20 ng/mL) for various treatment durations, and the phosphorylation of Rac1, p38, JNK, ERK1/2 MAPK, p65, and IκB NF-κB, AKT, and STAT3 was assessed. Experiments were performed in triplicate, and the representative blot is displayed. The band density was quantified and normalized to that of β-actin, which is presented as relative densitometric bar graphs. * *p* < 0.05, compared with the control group. Mann–Whitney U test, *n* = 3 (**B**) Effects of both NOX2- and NOX4-specific inhibitors on IL-1β-mediated chondrocyte signaling. IL-1β (20 ng/mL) was treated to primary chondrocytes with or without both NOX2-specific inhibitor (GSK2795039) (10 µM) and NOX4-specific inhibitor (GLX351322) (10 µM). Source blot data are provided in Appendix A.

**Figure 5 antioxidants-11-02346-f005:**
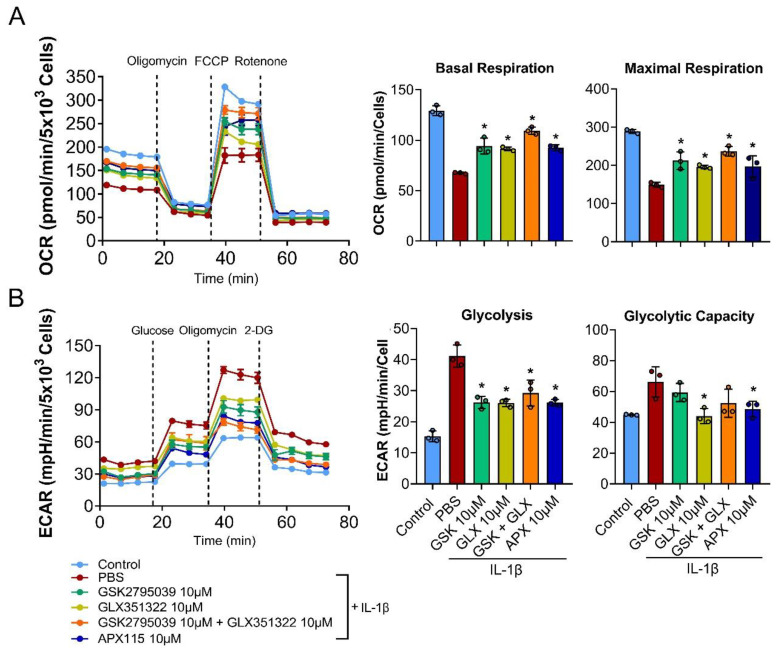
Effects of NOX inhibition on mitochondrial metabolism. (**A**,**B**) Primary chondrocytes were treated with IL-1β (10 ng/mL) with or without GSK2795039 (10 µM), GLX351322 (10 µM), and APX-115 (10 µM) for 24 h prior to the seahorse assay. (**A**) Oxygen consumption rate (OCR) measurement for the Mito stress test (sequential treatment of oligomycin, FCCP, and antimycin A/rotenone) and (**B**) extracellular acidification rate (ECAR) measurement for the glycolysis stress test (sequential treatment of glucose, oligomycin, and 2-DG) were performed on the Seahorse XF96 Extracellular Flux Analyzer. The basal and maximal respiration rates in the OCR analysis and the glycolysis and glycolytic capacity in the ECAR analysis from one time point was quantified and displayed as the mean ± SD. * *p* < 0.05 compared with the IL-1β-treated PBS control group. Mann–Whitney U test, *n* = 3.

**Figure 6 antioxidants-11-02346-f006:**
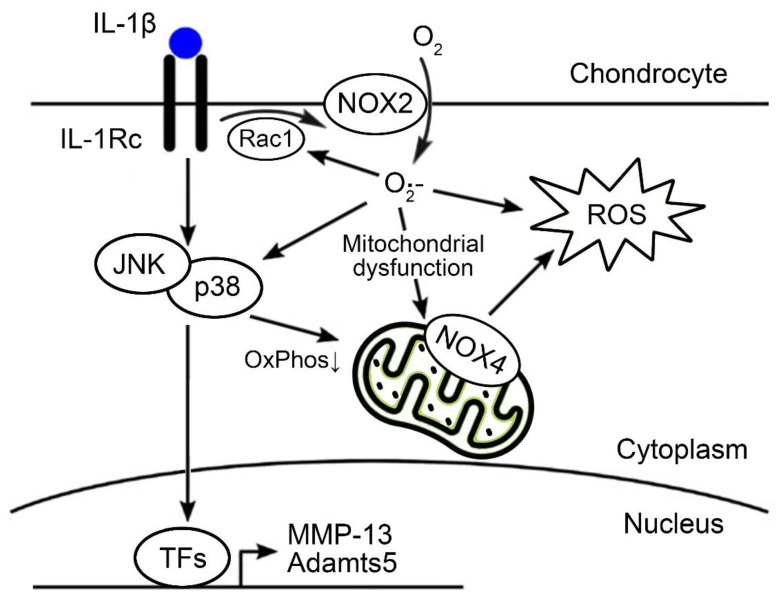
Scheme of the NOX system on OA chondrocytes evaluated in the present study. Among the NOX isoforms, NOX2 and NOX4 were dominantly expressed in OA chondrocytes. Functionally, NOX2- or NOX4-mediated ROS production may amplify the signal transduction of Rac1, p38, and JNK, and MAPK and directly or indirectly affect the dysregulation of mitochondrial metabolism. TFs, transcription factors; ROS, reactive oxygen species; NOX, NADPH oxidase.

## Data Availability

The datasets used and/or analyzed during the current study are available from the corresponding author upon reasonable request.

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
