# Peer review of "Inhibition of NADPH Oxidases Prevents the Development of Osteoarthritis"

_antioxidants, 2022, doi:10.3390/antiox11122346_

Round 1
Reviewer 1 Report (Previous Reviewer 1)
Satisfied with the author's reply. The reviewer may recommend for the publication. However, it will be better if the authors can get extra time for revision to check in vivo effect of both NOX2 and NOX4 KO mice on OA model.
Author Response
1-Please, carefully check definitions and use of abreviations: e.g. Interleukin-1B (76, 88, 159); 2-Deoxy-2-D-glucose (159, 257,389); FCCP (definition?)... Abbreviations should be defined the first time they appear in the text.
Author response: The overall use of abbreviations was checked again, and the full name of FCCP was added to line 222.
2-Please, use superscript for -ΔΔCt (180).
Author response: It was changed to 2−ΔΔCT.
3-Statistical analysis: Only Mann-Whitney test is mentioned but in legend to Figure 2A: One-way analysis of variance with Tukey's post test, as well as Mann-Whitney. It is not clear the method used. Besides, this non-parametric test is appropriate for two independent random samples but this does not seem to be the case.
It is very important to check the statistical analysis.
Author response: The legend of figure 2A about One-way ANOVA is actually a trace of the previous manuscript version. As the most of the in vivo and in vitro data were not normally distributed below 3-7/group of sample size, they were all analyzed using the non-parametric test, Mann-Whitney U test based on the reviewer's point about statistical method. Since non-parametric ANOVA test, the Kruskall Wallis test has no post-hoc analysis, the statistical significance between two groups were analyzed using Mann Whitney U-test, and these results were described in figures.
4-Page 8: "RNA" is mentioned several times, it should be mRNA.
Author response: I changed all the “RNA” in page 8 to “mRNA". Thank you for kind suggestion.
5-Legend to Figure 1: "invloved" (256), "mRNA expression are expressed" (265), "the relative mRNA expression of.. were presented..." (267).
Author response: I've corrected everything you pointed out: "invloved" (256) to “involved", "mRNA expression are expressed" (265) to “mRNA expression level was presented as ....", and "the relative mRNA expression of.. were presented..." to "the relative mRNA expression of.. was presented...".
6-Figure 3C: mRNA expression of "Arg" is shown but there is no mention in the text and there is no definition of "Arg". Is it aggrecan? Figure 3D and text mention "aggrecan". The presentation of results must be homogeneous.
Author response: "Arg" is a mistake, and we changed it to "aggrecan". The Figure 3C was has been revised. Thank you for kind suggestion.
7-Figure 4: there is no statistical analysis.
Author response: We added statistical analysis in revised manuscript in Figure 4C as bar graph with remarks of statistical significance. During revision, we found JNK signaling was also affected by NOX inhibition. We added JNK-related content throughout the manuscript.
8-Figure 4 and text: "Rac1 phosphorylation" is extensively used but Rac1 phosphorylation is not shown. Rac1 activation is correct.
Author response: I appreciate this precise point. I corrected the sentence as below; “Pan-NOX inhibition with APX-115 markedly suppressed the activated form of Rac1 (Rac1-GTP) and the phosphorylation of p38 MAPK...”
9-Legend to figure 4: "The phosphorylation of .... were assessed".
Author response: I've corrected everything you pointed out as below; "The phosphorylation of .... were assessed" to “the phosphorylation of .... was assssed”.
10-364:"phosphorylayed"
Author response: I changed it to “phosphorylated".
11-368 "... and catabolic proteases by IL-1B ..." ???
Author response: The catabolic proteases means MMP-13 and Adamts5. To clarify the meaning of the context, the sentence was changed as follows; “oxidative stress-marker genes and catabolic proteases such as MMP-13 and Adamts5 by IL-1β”
12-441: "regluatory"
Author response: I changed it to “regulatory".
13-456: "and in OA chondrocytes."
Author response: That means NOX1/3 was decreased by IL-1b in vitro and in articular cartilage of OA situation. To clarify the meaning of the context, the sentence was changed as follows; “NOX1 and NOX3 was down-regulated by IL-1β treatment in vitro and in chondrocytes of DMM model in vivo.”
14-485: "MAP kinases" MAPK is the correct abbreviation.
Author response: I changed it to “MAPK" in whole manuscript.
Reviewer 2 Report (New Reviewer)
The paper by Zhang and coworkers provides evidence of a prominent role of NOX2 and NOX4 in osteoarthritis (OA), and lays the basis for the use of NADPH-oxidase inhibitors in the therapy of this disease.
The authors used relevant in vitro and in vivo models, and appropriate methods to analyze the involved molecular pathways, and convincingly showed that NOX2 and NOX4 are upregulated in OA, are associated to ROS production, ECM degradation and mitochondrial function impairment. Furthermore, a pan-NOX inhibitor was able to counteract these effects both in vitro and in vivo.
Based on these consideration, the paper can be accepted in the present form for publication.
Author Response
1-Please, carefully check definitions and use of abreviations: e.g. Interleukin-1B (76, 88, 159); 2-Deoxy-2-D-glucose (159, 257,389); FCCP (definition?)... Abbreviations should be defined the first time they appear in the text.
Author response: The overall use of abbreviations was checked again, and the full name of FCCP was added to line 222.
2-Please, use superscript for -ΔΔCt (180).
Author response: It was changed to 2−ΔΔCT.
3-Statistical analysis: Only Mann-Whitney test is mentioned but in legend to Figure 2A: One-way analysis of variance with Tukey's post test, as well as Mann-Whitney. It is not clear the method used. Besides, this non-parametric test is appropriate for two independent random samples but this does not seem to be the case.
It is very important to check the statistical analysis.
Author response: The legend of figure 2A about One-way ANOVA is actually a trace of the previous manuscript version. As the most of the in vivo and in vitro data were not normally distributed below 3-7/group of sample size, they were all analyzed using the non-parametric test, Mann-Whitney U test based on the reviewer's point about statistical method. Since non-parametric ANOVA test, the Kruskall Wallis test has no post-hoc analysis, the statistical significance between two groups were analyzed using Mann Whitney U-test, and these results were described in figures.
4-Page 8: "RNA" is mentioned several times, it should be mRNA.
Author response: I changed all the “RNA” in page 8 to “mRNA". Thank you for kind suggestion.
5-Legend to Figure 1: "invloved" (256), "mRNA expression are expressed" (265), "the relative mRNA expression of.. were presented..." (267).
Author response: I've corrected everything you pointed out: "invloved" (256) to “involved", "mRNA expression are expressed" (265) to “mRNA expression level was presented as ....", and "the relative mRNA expression of.. were presented..." to "the relative mRNA expression of.. was presented...".
6-Figure 3C: mRNA expression of "Arg" is shown but there is no mention in the text and there is no definition of "Arg". Is it aggrecan? Figure 3D and text mention "aggrecan". The presentation of results must be homogeneous.
Author response: "Arg" is a mistake, and we changed it to "aggrecan". The Figure 3C was has been revised. Thank you for kind suggestion.
7-Figure 4: there is no statistical analysis.
Author response: We added statistical analysis in revised manuscript in Figure 4C as bar graph with remarks of statistical significance. During revision, we found JNK signaling was also affected by NOX inhibition. We added JNK-related content throughout the manuscript.
8-Figure 4 and text: "Rac1 phosphorylation" is extensively used but Rac1 phosphorylation is not shown. Rac1 activation is correct.
Author response: I appreciate this precise point. I corrected the sentence as below; “Pan-NOX inhibition with APX-115 markedly suppressed the activated form of Rac1 (Rac1-GTP) and the phosphorylation of p38 MAPK...”
9-Legend to figure 4: "The phosphorylation of .... were assessed".
Author response: I've corrected everything you pointed out as below; "The phosphorylation of .... were assessed" to “the phosphorylation of .... was assssed”.
10-364:"phosphorylayed"
Author response: I changed it to “phosphorylated".
11-368 "... and catabolic proteases by IL-1B ..." ???
Author response: The catabolic proteases means MMP-13 and Adamts5. To clarify the meaning of the context, the sentence was changed as follows; “oxidative stress-marker genes and catabolic proteases such as MMP-13 and Adamts5 by IL-1β”
12-441: "regluatory"
Author response: I changed it to “regulatory".
13-456: "and in OA chondrocytes."
Author response: That means NOX1/3 was decreased by IL-1b in vitro and in articular cartilage of OA situation. To clarify the meaning of the context, the sentence was changed as follows; “NOX1 and NOX3 was down-regulated by IL-1β treatment in vitro and in chondrocytes of DMM model in vivo.”
14-485: "MAP kinases" MAPK is the correct abbreviation.
Author response: I changed it to “MAPK" in whole manuscript.
This manuscript is a resubmission of an earlier submission. The following is a list of the peer review reports and author responses from that submission.
Round 1
Reviewer 1 Report
The manuscript entitled “Inhibition of NADPH oxidases prevent the development of osteoarthritis” is a promising study. However, The authors should address the following points to enrich their work for publishing in this high impact journal like Antioxidants.
- Introduction section is not well written. The NOX system is usually dormant in resting cells and needs stimuli for its activation. The authors should discuss how that happens in OA and why this is important for OA pathophysiology.
- The NOX system is very important other than producing oxidative stress. It plays a significant role in phagocytosis process which is important for the whole organism. This should also be discussed briefly in the introduction. Otherwise, it seems that NOX only plays negative effects by producing ROS.
- Therefore, the authors should also discuss the scope of global NOX inhibitor as it may interfere with other biological processes. It will be better to use KO mice line for different NOX isoforms to better understand which NOX isoform is the most important for OA disease progression.
- The authors should also comment (and include in the manuscript) about the particular ROS molecules (superoxide, hydroxyl radical etc.) generated in the system and explain its importance briefly.
- Describe the solubility of APX-115 (at 60 mg/kg) along with the DMSO % injected into control mice. Also, how did the authors find the dose and time for APX-115? Include this information in the manuscript.
- Supplementary figure 3. Full blots used in Figure 3D: the right lowest image for β-actin- is the molecular weight rightly described? Please explain.
- There are some grammatical and spelling mistakes in the manuscript. It is advised that the authors should rewrite the manuscript with the help of a native English speaker.
Author Response
The manuscript entitled “Inhibition of NADPH oxidases prevent the development of osteoarthritis” is a promising study. However, the authors should address the following points to enrich their work for publishing in this high impact journal like Antioxidants.
- Introduction section is not well written. The NOX system is usually dormant in resting cells and needs stimuli for its activation. The authors should discuss how that happens in OA and why this is important for OA pathophysiology.
Author comment: According to the reviewer’s comment, the introduction section was extensively revised. The comment about NOX activation in OA condition was described in line 87-94 as below.
“NOX system is dormant in resting cells and become activated when external stimuli trigger the NOX complex assembly consisted of two membrane subunits, gp91phox and p22phox and cytosolic subunits, p67phox, p47phox, p40phox and Rac2 (18). Although the activation mechanism of NOX system in the situation of OA is not fully understood, the possible candidate includes the inflammatory cytokines, such as interleukin (IL)-1β, IL-6 and heme oxygenase‐1, and the damage-associated molecular patterns (DAMPs) released during cartilage degradation, such as cartilage compositions, calcium crystals, and endogenous molecules (21, 22).”
- The NOX system is very important other than producing oxidative stress. It plays a significant role in phagocytosis process which is important for the whole organism. This should also be discussed briefly in the introduction. Otherwise, it seems that NOX only plays negative effects by producing ROS.
Author comment: In the revised introduction, we added the comment on what you pointed out in line 84-87. However, we think NOX system has both physiologic and pathologic role in chondrocytes, because the knockdown of NOX2 and NOX4 prevented endochondral bone formation (J Biol Chem. 2010;285(51):40294). So I briefly described it as below. Thank you for a valuable comment.
“NOX-derived ROS is functionally essential for bacterial killing and innate immunity (18). However, chondrocytes are non-phagocytic cells and the major role of NOX system in chondrocytes is for cell signaling and maintaining homeostasis (19, 20).”
- Therefore, the authors should also discuss the scope of global NOX inhibitor as it may interfere with other biological processes. It will be better to use KO mice line for different NOX isoforms to better understand which NOX isoform is the most important for OA disease progression.
Author comment: I totally agree with your opinion. It would be good to check in vivo effect of both NOX2 and NOX4 KO mice on OA model, but it could not be performed due to a short revision period of 10 days. However, the treatment of NOX2 and NOX4 inhibitors in vitro significantly reduced the expression of oxidative stress marker genes, as well as MMP13 and ADAMTS5. In addition, the expression of chondrogenic markers, Col2 and aggrecan was increased. Although it is a controlled in vitro data, I think this data reproduce the protective effects of NOX inhibitors in vivo. Further study on NOX2 or NOX4 conditional knockout mice is needed to better understand the role of NOX in OA.
- The authors should also comment (and include in the manuscript) about the particular ROS molecules (superoxide, hydroxyl radical etc.) generated in the system and explain its importance briefly.
Author comment: Thank you for valuable comment. Based on the reviewer’s comment, we added the comment on the particular ROS molecules in the situation of OA cartilage in line 58-64.
“The major types of ROS produced by chondrocytes are superoxide anion (O2∙) and nitric oxide (NO) (6). O2∙ is highly reactive that has a powerful influence on the intracellular redox state (6). It is catalyzed to H2O2 by superoxide dismutase and has very short cellular half-life of about 10-9s (7). Nitric oxide is synthesized by NO synthase (NOS), and chondrocytes express both endothelial NOS (eNOS) which is constitutively expressed, and inducible NOS (iNOS) which is induced by a lot of cytokines and growth factors (8).”
- Describe the solubility of APX-115 (at 60 mg/kg) along with the DMSO % injected into control mice. Also, how did the authors find the dose and time for APX-115? Include this information in the manuscript.
Author comment: The solubility of APX-115 in DMSO is 100 mg/ml. So we made 100 mg/ml of APX-115 stock solution with sonication. Then, to deliver 60 mg/kg APX-115, 18 μL of APX-115 stock solution was mixed with normal saline 342 μL and injected intra-peritoneally.
The pharmacokinetic data of APX-115 is not available until now. Therefore we decided the dosage and time for APX-115 based on the previous studies which have used daily oral administration at dose of 60 mg/kg APX-115 and showed clinically significant anti-inflammatory effects in diabetic mice (Lab Invest 2017;97(4):419-431, Free Radic Biol Med 2020;161:92-101). The bioavailability of small molecules measured by the area under the curve (AUC) revealed 1.5 to 6-fold higher after IP compared to oral administration (Pharm Res. 2019; 37(1): 12). We assumed that the bioavailability of IP is about 3-times higher than oral intake, and could get a similar AUC with oral intake if I gave APX-115 every 3-days. Based on these data, we gave 60 mg/kg APX-115 by IP twice a week to get a similar bioavailability with oral intake.
This information was added in the revised manuscript in line 129-137.
- Supplementary figure 3. Full blots used in Figure 3D: the right lowest image for β-actin- is the molecular weight rightly described? Please explain.
Author comment: The molecular weight of beta-actin is 42 kDa. The size ladder of the right lowest image for beta-actin of supplementary figure 3 was marked incorrectly as it was pushed back one by one. Actually, beta-actin is located between 35 kDa and 55 kDa. We have corrected it during revision.
- There are some grammatical and spelling mistakes in the manuscript. It is advised that the authors should rewrite the manuscript with the help of a native English speaker.
Author comment: This manuscript has already received a commercial English editing service before this submission. The period given to this revision is 10 days, which is relatively short to correct the language. Of course, I will request the English correction service provided by the Antioxidant after decision.

Reviewer 2 Report
The manuscript proposed by Han and colleagues is aimed at evaluating the role of NADPH oxidase in cell and animal models of osteoarthritis (OA). The main results demonstrated that IL-1beta, one of the main proinflammatory interleukines involved in OA, is accompanied by the increase of NOX2 and NOX4 isoforms and by enhanced oxidative stress. Importantly, the inhibition of NOX isoforms counteracted OA though the inhibition of catabolic signals and the restoration of mitochondrial functionality. When evaluated a s awhole, the work is well written, the experimental flux is coherent and the methodological approaches are nicely designed, as well as statistics. Results are convincing and sufficiently integrated with the state of the art in the discussion section.
There are only few shortcomings I would like to highlight:
1) Sometimes, the co-administration of both the NOX2 and NOX4 inhibitors does not mediate an additive effect. In addition, the administration of the pan-NOX inhibitor APX115 seems to be more efficient if compared to the GSK/GLX combination, suggesting that the inhibition of NOX2 and NOX4 only partially explain the effects mediated by APX115. For instance, this is the case for some targets evaluated in figure 3C and D. The authors should provide an explanation.
2) Data are expressed as the mean ± S.E.M. The standard error of the mean indicates the uncertainty of how the sample mean represents the population mean. In my opinion, the authors inappropriately report the SEM instead of the Standard Deviation (SD). Since the SEM is always less than the SD, it deceives the reader into underestimating the variability between individuals within the study sample.
3) NADPH oxidase activity is strictly dependent on Rac1 recruitment on cell membrane. This should be evaluated.
4) Concerning the working model (figure 6), the connection between IL-1beta and NADPH oxidase is not evident.
Author Response
The manuscript proposed by Han and colleagues is aimed at evaluating the role of NADPH oxidase in cell and animal models of osteoarthritis (OA). The main results demonstrated that IL-1beta, one of the main proinflammatory interleukines involved in OA, is accompanied by the increase of NOX2 and NOX4 isoforms and by enhanced oxidative stress. Importantly, the inhibition of NOX isoforms counteracted OA though the inhibition of catabolic signals and the restoration of mitochondrial functionality. When evaluated as a whole, the work is well written, the experimental flux is coherent and the methodological approaches are nicely designed, as well as statistics. Results are convincing and sufficiently integrated with the state of the art in the discussion section.
There are only few shortcomings I would like to highlight:
1) Sometimes, the co-administration of both the NOX2 and NOX4 inhibitors does not mediate an additive effect. In addition, the administration of the pan-NOX inhibitor APX115 seems to be more efficient if compared to the GSK/GLX combination, suggesting that the inhibition of NOX2 and NOX4 only partially explain the effects mediated by APX115. For instance, this is the case for some targets evaluated in figure 3C and D. The authors should provide an explanation.
Author comment: I agree with the reviewer’s opinion. Actually NOX1 and NOX3 is also expressed in unstimulated primary chondrocyte and decreased by catabolic stimuli such as IL-1b treatment (Figure 2A, B). The partial expression of NOX1 and NOX3 may be the reason that the pan-NOX inhibitor APX115 is more potent than GSK/GLX combination. Another meaning or this data is that NOX1 and NOX3 can play an important role in physiological conditions such as endochondral bone formation or differentiation of mesenchymal cells into chondrocytes. The comment on this is described in line 487-491.
In Figure 3 C, D, although the GSK/GLX combination is not significantly effective in RNA expression compared to single treatment, it is comparable to that of APX-115. And the combination of NOX2 and NOX4 inhibitor showed an additive effect, especially in Figure 3E (Adamts5, Col2 WB) and Figure 5 (chondrocyte metabolism).
2) Data are expressed as the mean ± S.E.M. The standard error of the mean indicates the uncertainty of how the sample mean represents the population mean. In my opinion, the authors inappropriately report the SEM instead of the Standard Deviation (SD). Since the SEM is always less than the SD, it deceives the reader into underestimating the variability between individuals within the study sample.
Author comment: I agree with your opinion. We changed all the statistics of SEM to SD in the process of revision. Thank you for an appropriate suggestion.
3) NADPH oxidase activity is strictly dependent on Rac1 recruitment on cell membrane. This should be evaluated.
Author comment: Rho-like small GTPases Rac1 or Rac2 is critical for the activation of NOX family, especially IL-1b-mediated ROS production. It would be good to check the activity of Rac1 and Rac2 in figure 4. However, we have very short time for revision to confirm the Western blot data 3 times. In addition, I think that the activation of Rac protein in chondrocytes has another significant meaning in terms of signaling mechanisms involving NOX activation, and thus, is seems to be beyond the scope of this study.
4) Concerning the working model (figure 6), the connection between IL-1beta and NADPH oxidase is not evident.
Author comment: Based on the reviewer’s comment, we modified the figure 6 depicting that IL-1b signaling activates NOX2 through Rac.

Round 2
Reviewer 2 Report
I carefully read the response letter provided by the authors. I continue to have concerns about 4. In the new version of the working model, the authors suggest that NADPH oxidase is influenced by IL-1beta by the activation of Rac. However, the authors did not provide any evidence for Il-1beta-mediated Rac activation (please note that I asked for Rac analysis in the previous round of review). Additionally, the authors demonstrated that IL-1beta stimulation led to an increased expression of NOX2 and NOX4 subunits, suggesting long-term mechanisms of regulation instead of short-term activation by Rac recruitment.
In conclusion, the potential role of Rac still remains elusive, a putative mechanistic explanation linking IL-1beta and NADPH oxidase is not provided and the working model does not adequately reflect the results shown in this work. These aspects should be clearly discussed and addressed in the manuscript. For instance, the lack of Rac estimation should be integrated in the main text as a limitation of this study. Putative mechanisms linking IL-1beta administration and NOX2-4 increased expression should be at least speculated. These points are critical to explain the model proposed in Figure 6. As a consequence, figure 6 should be properly amended.